# The Drought Regime in Southern Africa: A Systematic Review

**Fernando Maliti Chivangulula** [1] , **Malik Amraoui** [1] **and Mário Gonzalez Pereira** [1,2,*]

1   Centre for Research and Technology of Agro-Environmental and Biological Sciences (CITAB), Inov4Agro, University of Trás-os-Montes and Alto Douro (UTAD), Quinta de Prados, 5000-801 Vila Real, Portugal; al75113@alunos.utad.pt (F.M.C.); malik@utad.pt (M.A.)

2   Instituto Dom Luiz (IDL), FCUL, Campo Grande Edifício C1, Piso 1, 1749-016 Lisboa, Portugal

*   Correspondence: gpereira@utad.pt; Tel.: +351-259-350-728

**Abstract:** Drought is one natural disaster with the greatest impact worldwide. Southern Africa (SA) is susceptible and vulnerable to drought due to its type of climate. In the last four decades, droughts have occurred more frequently, with increasing intensity and impacts on ecosystems, agriculture, and health. The work consists of a systematic literature review on the drought regime's characteristics in the SA under current and future climatic conditions, conducted on the Web of Science and Scopus platforms, using the PRISMA2020 methodology, with usual and appropriate inclusion and exclusion criteria to minimize/eliminate the risk of bias, which lead to 53 documents published after the year 1987. The number of publications on the drought regime in SA is still very small. The country with the most drought situations studied is South Africa, and the countries with fewer studies are Angola and Namibia. The analysis revealed that the main driver of drought in SA is the ocean–atmosphere interactions, including the El Niño Southern Oscillation. The documents used drought indices, evaluating drought descriptors for some regions, but it was not possible to identify one publication that reports the complete study of the drought regime, including the spatial and temporal distribution of all drought descriptors in SA.

**Keywords:** drought regime; Southern Africa; drought factors; drought impacts; drought indices

## 1. Introduction

Drought can be defined as a period of abnormally dry weather long enough to cause a significant hydrological imbalance [1]. Drought is a relative term, in the sense that the rainfall deficit must be evaluated taking into account the climatological normality of precipitation and all precipitation/water-related activities, also because the local hydrological imbalance depends on the supply, demand, and management of water, which reflects the role of the hardly quantifiable human interference [2]. Therefore, drought cannot be defined as a purely natural phenomenon that occurs when precipitation is significantly lower than the climatological normal [3,4], which also depends on the study period, especially in the non-stationary context of climate change [2].

As opposed to traditional and conventional definitions of drought based on the deficit in water-dependent variables or activities (for example, precipitation, soil moisture, surface and groundwater storage and irrigation) associated with natural climate variability, more recently, the definition of anthropogenic drought, within the scope of the coupling of climate-water systems, which understands drought as a process and not a product to better frame and describe the complex and interrelated dynamics of natural and human-induced conditions and changes [5]. In this definition, drought includes the entire spectrum of dynamic processes that are not necessarily linear in human–nature systems (e.g., earth–atmosphere interactions, water and energy balance), which explains drought as a composite multidimensional and multiscale phenomenon governed by climate variability and change, natural variability of the water cycle, human decisions and activities, including land and

water management. Drought is part of the climate, occurs in any type of climate, and is not restricted to periods of abundant or scarce precipitation [6,7].

One of the drought's complexities is quantifying its associated impacts [8]. For example, a precipitation deficit during the plant growth phase affects agricultural production, water supply systems, groundwater storage, and changes in soil moisture conservation [1]. In this sense, traditionally, droughts can be classified into four types of droughts: Meteorological drought, hydrological drought, agricultural drought, and socioeconomic drought [2,9,10].

Meteorological drought is characterized by a deficit of precipitation in historical records and depends on the type of local climate [4]. Meteorological drought refers to the deficiency of precipitation, can be related to the increase in potential evapotranspiration in a given temporal space [11], and normally affects the ecological patterns and processes of terrestrial ecosystems [12]. Agricultural drought, or simply, soil moisture drought [1], refers to a deficit of soil moisture or a decrease in the amount of water available in the different soil layers [13]. During the agricultural drought, there is a reduction in the supply of soil moisture to the vegetation [11].

Hydrological drought describes a period with a significant decrease in the normal levels of surface and underground water resources or negative anomalies in groundwater flow levels. Hydrological drought is characterized by the occurrence of long periods of significant rainfall deficit able to lead to inadequate surface and subsurface water resources for established water uses of a given water resources management system and is observed after meteorological and agricultural droughts [9,13]. Concretely, hydrological drought refers to the significant reduction in the amount of water in the hydrological system, namely in abnormally low flows in rivers and abnormally low levels in lakes, reservoirs, and groundwater [11]. Socioeconomic drought is associated with the impossibility of water resource management systems to meet human needs or even the lack of water to meet the water needs of populations [6,9,14].

Operational and customary drought is assessed with different drought indices. Series of drought indices are time series of numerical values that allow us to evaluate the start and end dates, duration, and other characteristics of droughts [9]. The World Meteorological Organization (WMO) categorizes and classifies drought indices based on the type of data used (e.g., data observed in situ or by remote sensing, climatic or hydro-meteorological elements, or parameters) as being meteorological, soil moisture, hydrological, remote sensing, as well as composite or modeled indices [4]. While there are several drought indices, there is no index that is globally and adequately applicable in all cases. The most well-known and widely used drought indices around the world are the Palmer Drought Severity Index (PDSI), Standardized Precipitation Index (SPI), and Standardized Precipitation Evapotranspiration Index (SPEI) [15,16].

The PDSI is typically used to measure very long droughts and is calculated from monthly precipitation and surface air temperature data to estimate soil moisture supply and demand [9,17]. PDSI is used to measure land surface aridity anomalies, which are correlated with soil moisture and land water accumulation variations [18]. The SPI is a drought index calculated based on precipitation alone and can identify drought across multiple time scales [16,19]. The SPEI is calculated with precipitation and air temperature data, it is considered an improvement of the SPI to also take into account the evapotranspiration process in addition to the precipitation deficit and can also be calculated on several time scales [10,20]. These three standardized indices have the advantage that they can be acquired and compared for different locations and periods [21].

The vegetation indices can also be used to assess drought based on its effects on vegetation. The most commonly used vegetation indices to assess drought are the Normalized Difference Vegetation Index (NDVI), Enhanced Vegetation Index (EVI), and Vegetation Condition Index (VCI) [4]. The NDVI is a plant index that is based on plants' reflectance visible and near-infrared wavelength bands of the electromagnetic spectrum and is used for identifying and monitoring droughts affecting agriculture [22]. The EVI allows the

identification of plant water stress associated with drought. The VCI is a vegetation index fine-tuned to quantitatively and qualitatively determine the impact of drought on vegetation and provides details linked to terrestrial ecological conditions, and is widely applied in agriculture [23].

Drought indices are one of the most common tools to assess the occurrence and effects of drought, as well as different drought descriptors or parameters, which involve number, frequency, duration, intensity, severity, and spatial extent [9]. For example, meteorological drought indices allow for an operational definition of drought and its characteristics. These indices are computed for specific time scales, i.e., precipitation frequencies at time scales of 1, 3, 6, 12, 24, 36, and 48 months [24–26]. Drought over this range of time scales is associated with a specific type of drought. For example, 1- to 2-month SPI drought is indicative of meteorological drought, 1- to 6-month SPI corresponds to agricultural drought, and 6-month to longer scales SPI can be used to assess hydrological and socioeconomic drought [6,27].

The analysis of the time series of each drought index obtained for each time scale allow us to evaluate the occurrence and characteristics of the drought at that time scale [26]. For example, in the case of PDSI, SPI, and SPEI, a drought starts when the index assumes a negative value and ends when the index back to assume positive values (Table 1). The duration of a drought event is simply the difference between the end and start date [28,29], i.e., the period of consecutive months of drought is considered a drought event [10]. Usually, drought is assessed on a monthly scale, and each event lasts at least two to three months but can extend to several months or years [30]. Drought frequency can be defined by the number of months of drought that occurred in all months during 30 years [10] or by the number of drought events divided by the duration of the study period [24]. Drought severity is determined by the absolute value of the sum of the index values during a drought episode [24]. Drought intensity is the average value of the drought index below the climatic normal and is determined by dividing the drought severity by the duration [9,24]. Some authors link the severity of the drought to the duration, intensity, and spatial extent of the occurrence of a specific drought event, as well as to the impacts on ecosystems due to lack of water, being more severe the more negative the drought index values are (Table 1) [30].

**Table 1.** Drought classification criteria of PDSI, SPI, and SPEI [31].

| Drought Class | PDSI Value | SPI and SPEI Value |
|---|---|---|
| Extremely wet | $\geq$4.00 | $\geq$2.00 |
| Severely wet | 3.00 to 3.99 | 1.50 to 1.99 |
| Moderately wet | 2.00 to 2.99 | 1.00 to 1.49 |
| Slightly wet | 1.00 to 1.99 | 0.50 to 0.99 |
| Near Normal | −0.99 to 0.99 | −0.49 to 0.49 |
| Mild dry | −1.99 to −1.00 | −0.99 to −0.50 |
| Moderate dry | −2.99 to −2.00 | −1.49 to −1.00 |
| Severe dry | −3.99 to −3.00 | −1.99 to −1.50 |
| Extremely dry | $\leq$−4.00 | $\leq$−2.00 |

The impacts of drought vary according to the type of drought. For example, meteorological and soil moisture droughts affect agriculture, terrestrial, and aquatic ecosystems. On the other hand, hydrological drought affects several systems, from the reduction of the amount of water for agriculture and human consumption. It also affects ecosystems, energy production, and industry [11]. Due to the size of the impacts associated with drought, this is thus considered one of the most costly natural disasters with significant impacts widespread worldwide and particularly in Africa [3,4]. Worldwide and in the period from 1970 to 2019, droughts accounted for 6% of the total number of natural disasters, accounted for 7% of reported economic losses, were responsible for 34% of all human deaths caused by natural disasters, which makes droughts the second natural disaster responsible for the highest number of human deaths, after tropical cyclones. For Africa, these results are

even more significant, given that 15% of natural disasters are related to climate and water and 35% of associated deaths [32]. For example, in Southern Africa (SA) the drought of 1991–1992 affected more than 20 million people [30].

As drought is one of the natural disasters with high socioeconomic costs [33], understanding its characteristics and how they evolve is essential to improve our ability to plan the management of water resources. Based on the scientific literature, many studies of systematic reviews on drought have been carried out globally, but through a random search in Web of Science and Scopus databases, it was possible to identify a few studies for some African countries and regions, e.g., [9,34–41], but rarely for entire SA. Therefore, to assess and fill this gap shortly, the present study aims to evaluate the current state of knowledge about the characteristics of the drought regime in SA, under current and future climate conditions. Consequently, this study will also identify knowledge gaps about the drought regime in SA. Specifically, our study seeks to answer the following research questions (RQ):

RQ1 What are the characteristics of the drought regime in the current climatic conditions? In the context of this RQ1, it is hypothesized that: (i) At least some of the drought descriptors in SA are known, (ii) at least some of the spatial and temporal patterns of the drought regime in SA are known. Additionally, the main objective is to characterize the drought regime in SA under current climate conditions.

RQ2 What are the main materials and methods used to characterize the drought regime in SA? In the framework of the RQ2 the hypotheses were the following: (i) Most of the previous studies relied on drought and/or vegetation indices, (ii) most of the previous studies were based on remote sensing and ground data, (iii) most of the previous studies used physical and/or statistical methods/models to characterize the drought regime. Within RQ2, the objective was to assess the methodological approaches used to characterize the drought regime in SA.

RQ3 What are the main factors and impacts of drought in SA? Within the scope of this RQ3, the research hypotheses were: (i) The El Niño/Southern Oscillation (ENSO) is likely to be one of the main drivers of drought in SA, (ii) ocean circulation and sea surface temperature (SST) patterns in the Atlantic and Indian Oceans play an important role in the occurrence and character of droughts, (iii) some of the atmospheric circulation patterns in the region can influence the precipitation regime and, consequently, the drought regime, and (iv) the list of effects of drought in SA will include all the socioeconomic consequences resulting directly and indirectly from water scarcity in natural and human systems, namely in agricultural, hydroelectric, industrial, health, hunger, and human mortality. In the context of RQ3, the objective was to better understand the main driver and consequences of drought in SA.

RQ4 What are the existing projections on the drought regime in SA for different periods and scenarios of future climate? In this case, our hypothesis included: (i) There are already some drought regime projections for the SA, (ii) namely, for some future periods and scenarios, but (iii) there should not yet be estimates for the new climate scenarios or the entire territory of SA.

Studies or review articles are excellent elements of study for researchers, especially the younger ones and those starting research on a new topic. In this sense, this study also has the secondary aim of gathering and providing a set of fundamental information on the drought regime in general and in Southern Africa in particular.

## 2. Materials and Methods

This review was conducted systematically using the structure of the flowchart (Figure 1) of the Preferred Reporting Items for Systematic Reviews and Meta-Analysis protocol (PRISMA) and consisted of (1) identifying the bibliographic databases, (2) defining the research equation, (3) selecting the selected documents through criteria, and (4) evaluating the quality of the studies and synthesizing the information [42]. The development of the search strategy to identify the studies about the drought regime in SA was carried out in

March and April 2023, on the Web of Science (WoS) and Scopus platforms, complemented by the Google Scholar search engine and using a search equation in the title of the articles:

$$\text{Drought * AND Southern AND Africa *} \tag{1}$$

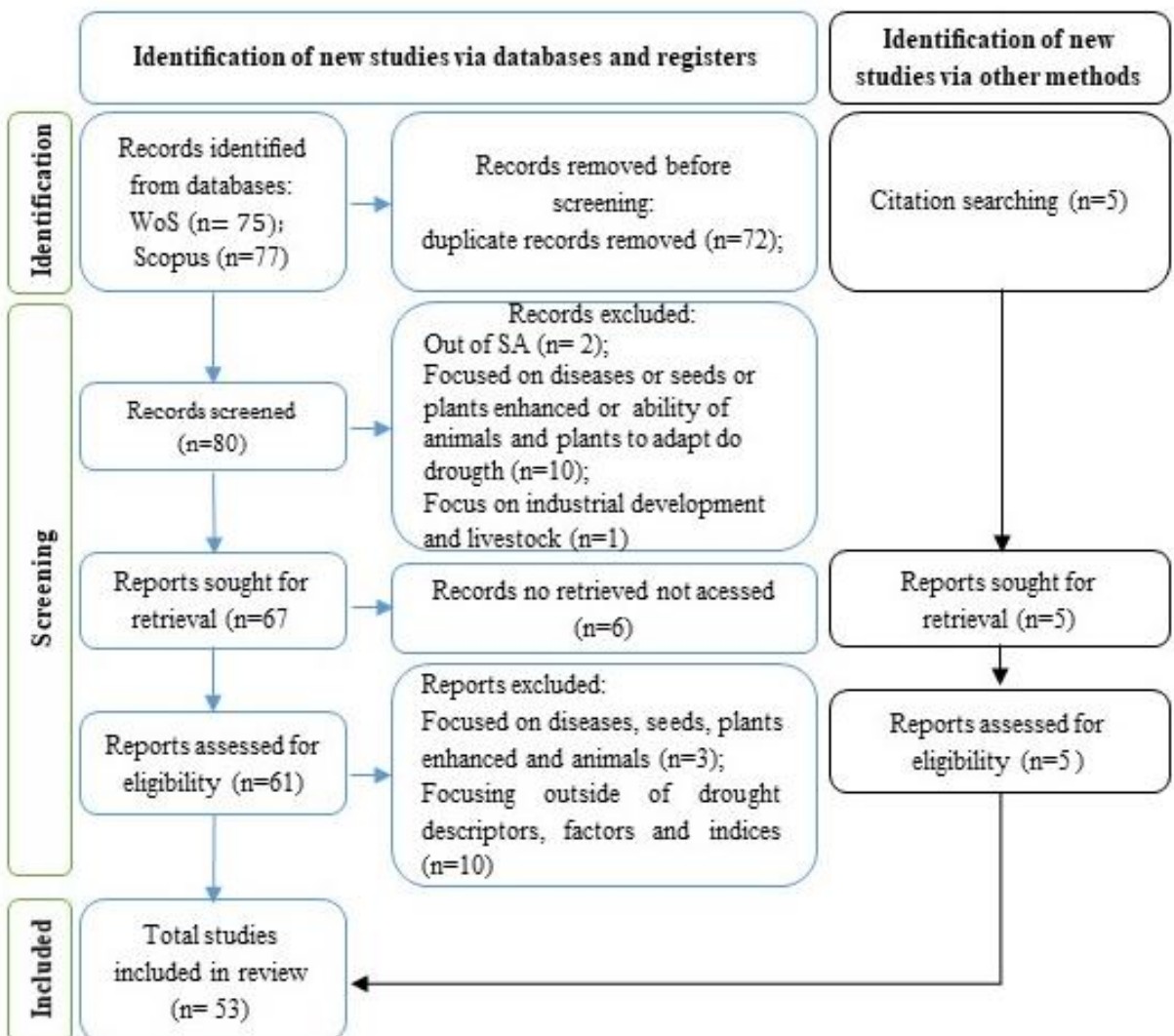

**Figure 1.** PRISMA 2020 flowchart of this systematic literature review. Adapted from [42].

The equation made it possible to identify all studies that include, in the title, the keywords indicated in the research equation above. We decide to use the wildcard characters (*) in the keywords Drought and Africa to be able to find other nearby words like Droughts and African. To achieve the research objectives, several inclusion and exclusion criteria were also adopted/defined (Table 2). It is important to point out that the inclusion and exclusion criteria were defined with two main objectives: (i) To achieve the main goal/aim of identifying a complete set of documents published since 1987, accessible to the generality of the scientific community, on the drought regime in southern Africa, under current and future regime conditions, and (ii) ensure that they did not contribute to increasing the risk of bias.

**Table 2.** Criteria for inclusion and exclusion of documents in the systematic search.

| Inclusion Criteria | Exclusion Criteria |
| --- | --- |
| Written in English. | Not be written in English. |
| Peer-reviewed articles and journals. | Non-peer-reviewed articles and journals. |
| Focus on the drought descriptors, factors, and impacts. | Not focused on drought descriptors, factors and impacts. |
| WMO Reports. | Documents not published by WMO and indexed journals. |
|  | Study area outside SA. |

The PRISMA2020 methodology defines three fundamental steps for carrying out a systematic review: (i) Identification, (ii) screening, and (iii) inclusion (Figure 1). The first step consisted of identifying the publications from the two databases and the exclusion of duplicates through the screening carried out in the Mendeley Desktop reference management and human intervention. The second step included three stages: The first stage included the reading and analysis of the title, abstract, and keywords of each publication, which may lead to excluding an additional number of publications. In the second stage, the inaccessible publications were also excluded from the list. Finally, in the third stage of step two, the publications were completely read, and the inclusion and exclusion criteria (Table 2) were used to further keep or reject publications.

The application of these three steps led to the final list of publications that served as the basis for the systematic literature review on the drought regime in SA. This entire process that led to the final list of publications was carried out based on the collection and analysis in Microsoft Excel of qualitative and quantitative information extracted from the publications throughout the different steps. The Excel spreadsheet allowed the inclusion and organization of information on the different topics of the drought regime, including drought descriptors, current and future patterns, factors/drivers, consequences/impacts, input data, and methodology (use of drought indices and other methods).

## 3. Results and Discussion

### 3.1. Publications Identified with the PRISMA2020 Methodology

The application of the PRISMA2020 methodology allowed us to identify a total of 152 publications via databases and registers, as well as five publications via other methods, namely citation searching (Figure 1). All the identified publications in databases can be accessed at the website address presented in Table 3. The publication identification via databases included review studies and a very small number of studies about drought carried out in Angola and Namibia. This motivated the inclusion of these five publications identified by citation searching for review articles and presenting some useful information on the drought regime in these two countries. One of these publications is a case study for southern Angola performed to evaluate drought risk in data-scarce contexts [43], while another is about the use of the Blended Drought Index to assess the integrated drought hazard [44].

All identified studies in the search were entered into Mendeley Desktop reference management software and reconfirmed by human verification. In the first step, 72 duplicate publications were excluded which corresponds to 47% of the total number (152) of identified publications via databases. In the second step, 12.5% of the publications were excluded due to: (i) Study area out of SA (1.3%), (ii) focused only on diseases, improved seeds or plants, or the ability of animals and plants to adapt to drought (6.6%), focused on industrial development or livestock (0.66%), and inaccessible documents (3.9%). In the third step, 8.6% of publications were excluded for not including drought descriptors, factors, or indices (6.6%) and focused only on drought impacts on diseases, seeds, plants, and animals (2.0%).

**Table 3.** Bibliographic databases, website address where the output of the search is saved, number of publications identified (N), access date and search procedure to identify the publications to perform the systematic review.

| Database | Website Address | N | Access Date | Search Procedure |
|---|---|---|---|---|
| WoS | https://www.webofscience.com/ | 75 | 14 March 2023 | Research equation |
| SCOPUS | https://www.scopus.com/ | 77 | 14 March 2023 | Research equation |
| Google Scholar | https://www.mdpi.com/2225-1154/5/3/51 | 1 | 22 April 2023 | Citation searching |
| | https://iwaponline.com/jwcc/article/ | 1 | 22 April 2023 | Citation searching |
| | https://centaur.reading.ac.uk/108297/ | 1 | 28 June 2023 | Citation searching |
| | https://www.mdpi.com/2073-445X/11/2/159 | 1 | 28 June 2023 | Citation searching |
| | https://www.mdpi.com/1660-4601/19/9/5082 | 1 | 28 June 2023 | Citation searching |

Of the 157 publications initially identified in the databases, records, and citation search, 85 remained after removing repetitions and 53 after applying the inclusion/exclusion criteria. The 85 documents were published between 1987 and 2023, which corresponds to an average of 2.4 publications per year, but 86% of these documents were published from 2000 onwards (Figure 2). The 53 documents selected for the literature review were published after 2000, which corresponds to an average of 1.5 publications per year (Figure 2). However, 68% of the documents included in the literature review were published in the last decade, and 2020 was the year in which the largest number of these documents was published (11%). This growing trend suggests/demonstrates that this subject has gained interest in recent years by the scientific community, eventually associated with the frequency and magnitude of drought impacts in SA. However, the number of publications on the drought regime in SA is still very small, given the large size of its territory.

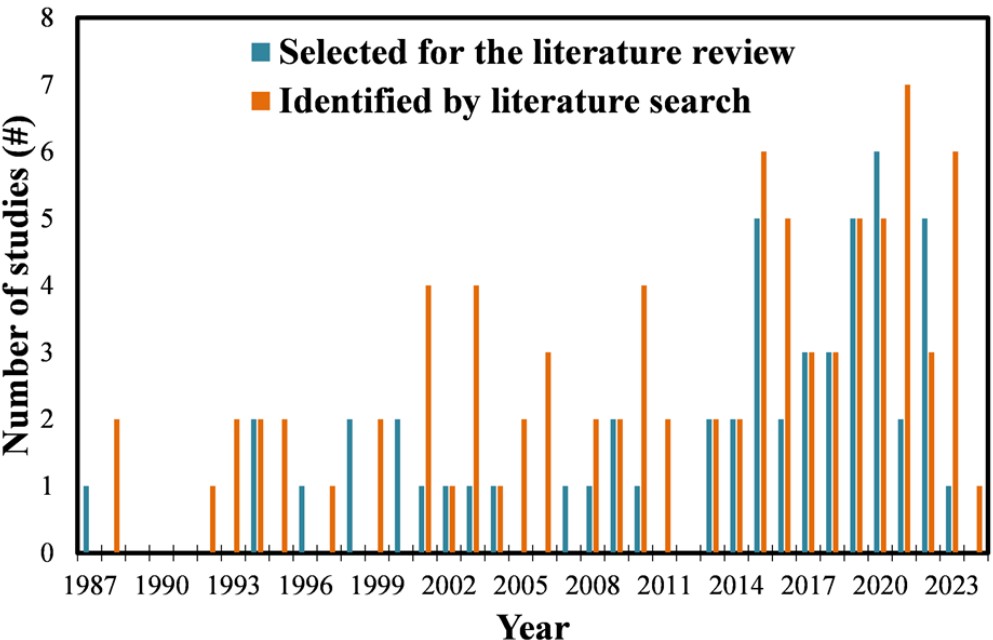

**Figure 2.** Annual numbers of documents identified by literature search (1987–2023) and selected/included for the literature review.

Although the bibliographical research was carried out to identify documents whose study region was SA, only 34% of the 85 documents refer to studies in this region, that is, most documents report/describe studies carried out in a region, part or all of it of a country or group of countries of SA (Figure 3). Most of these studies accounted for across SA are restricted, for example, to a drought event, to one or two drought descriptors, or to some drought classes and time scales. In addition, 14% of these documents include

studies aimed at minimizing the social and economic impacts of drought, the creation of monitoring tools, drought warning systems, or even improvements in seeds and crops to adapt them to drought.

An analysis of the study areas of the 85 documents identified in the databases, registries, and citation searches (Figure 3) reveals that the countries where the most studies were carried out were South Africa 31 (36%), Zimbabwe 20 (24%), and Mozambique 18 (21%), while the countries where drought was less studied were Eswatini (formerly Swaziland) 9 (11%), Namibia 4 (5%), and Angola 3 (4%). It is important to highlight that the study area of each document can be more than one country, which explains the sum of percentages greater than 100%. Regarding the 53 documents selected for the literature review, the distribution is essentially similar, with an increase of documents for SA (by 4%), South Africa (7%), and Angola (2%). These results disclose the need to acquire and/or update knowledge about drought throughout the SA, as none of the documents fully describe the drought regime in this region. This need is particularly evident and urgent for SA countries for which there is a greater lack of more and better information about this natural disaster.

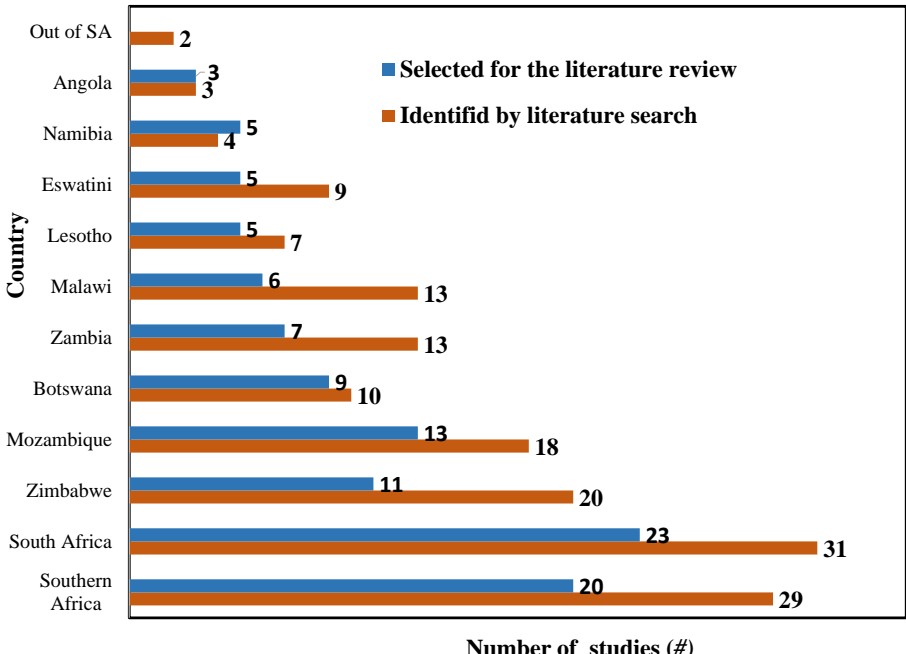

**Figure 3.** The number of publications related to each country of the study region (Southern Africa) identified by the literature search and selected/included for the literature review (1987–2023).

### *3.2. Literature Review*

From this section, the statistics presented, unless otherwise indicated, are related to the 53 articles and book chapters selected for the literature review.

### 3.2.1. Drought Factors

About half (24 in 53, 45%) of the documents included in the literature review studied drought factors or drivers in SA (Table 4). It is important to highlight that each document can study/discuss more than one factor. The document analysis revealed that the most important drought drivers in SA are (i) Ocean–Atmosphere interactions, with 50% of the total number of studies included, comprising El Niño Southern Oscillation (ENSO), other Ocean–Atmosphere interaction events, namely between the Indian and/or Atlantic Ocean and the Atmosphere, SST anomalies, (ii) anthropic influence, including fires, gas emissions, and global warming (13%), (iii) heat waves (4%) and (vi) lack of preparation and follow-up (2%), which are important factors of the magnitude of the drought impacts.

**Table 4.** Main drought factors in SA. The absolute (N) and relative (n) number of citations are also provided.

| Factors | Studies | N (#) | n (%) |
|---|---|---|---|
| El Niño Southern Oscillation (ENSO) | [29,45–57] | 14 | 26 |
| Other Ocean–Atmosphere interactions | [33,48,51,54,56–58] | 7 | 13 |
| Sea surface temperature (SST) | [47,48,50,51,55,59] | 6 | 11 |
| Anthropic influence (e.g., fires, gas emissions, and global warming) | [24,33,52,60–63] | 7 | 13 |
| Heatwaves | [61,64] | 2 | 4 |

The results of the literature review about the drought factors (Table 4) point to ENSO as the main driver of the most severe drought events and source of climate variability and predictability in the SA region [45,49]. ENSO plays a crucial role in defining drought in SA, associated with more than 66% of the severe droughts that occurred in the region, e.g., [45–47,53–57]. For example, 8 of the 12 most severe droughts at 6 and 24 months in the 1901–1999 period were also ENSO years [45].

ENSO is the combination of El Niño (EN) and Southern Oscillation (SO). El Niño is characterized by a positive and significant anomaly of SST in the equatorial Pacific Ocean near South America. The Southern Oscillation is an interannual fluctuation of atmospheric pressure at sea level over the Pacific Ocean, usually evaluated with the Southern Oscillation Index (SOI), which is a standardized index based on the sea level pressure (SLP) differences between Tahiti and Darwin, Australia [48,55,65].

ENSO is associated with a vast set of significant anomalies in different climatic elements (e.g., temperature, precipitation) and linked to the occurrence of extreme events and natural disasters (e.g., wildfires, droughts, floods) observed all over the world through a process usually called teleconnection. Teleconnections are relationships between atmospheric disturbances or the Ocean–Atmosphere interaction with the climate in distant regions of the globe. Usually (neutral or La Niña conditions), SST is characterized by lower values at the eastern edge and higher values at the western edge of the equatorial Pacific, while a Walker cell circulation is observed in the atmosphere. In this case, precipitation is particularly high over the western edge of the Pacific Ocean as a result of convection promoted by strong heating and evaporation in that region [45,48,51,57].

Under El Niño conditions, an inversion of the SST pattern is observed, an Eastward shift of the Walker cell, and, consequently, of the region of high precipitation. The large-scale heating (El Niño) or cooling (La Niña) of the equatorial Pacific SST affects lower tropospheric pressure fields and alters the Walker circulation, which, in turn, affects the transport of moisture, causing excess or deficit of precipitation in different regions [57]. These climate anomalies also include variability within the Indian and Atlantic Oceans [57]. El Niño years are associated with generalized dry conditions during summer, being stronger in the southeastern part of SA, while La Niña years are more favorable to rainy conditions or high precipitation in many regions of the globe [57,65]. Additionally, anthropogenic climate change will contribute substantially to a significant increase in the severity and frequency of droughts in SA as anthropogenic warming will significantly contribute to the increase of the El Niño events and increase the likelihood of changing drought characteristics in the region [29,61,62].

Events similar to El Niño are also observed in the South Atlantic coasts of Angola and Namibia, where an event called Benguela Niños is usually caused by a specific wind stress events in the equatorial Atlantic Ocean, which triggers the occurrence of warm and extreme events in the region [57]. Benguela Niños are associated with floods in Angola and Namibia, heavy precipitation events in the arid Namib desert as well as frequent droughts in the semi-arid region of SA, namely in the Benguela region of Angola [57].

SST/Ocean–Atmosphere interactions are not the only driver of drought in SA, as 20% of the studies also point to other factors [33,47,48,51,60,64]. Drought is caused by below-normal rainfall but also by changes in the hydrological year, i.e., delay of the beginning

or anticipation of the end of the rainy season [45]. For example, the severe drought in 2002–2003 was caused by a rainfall deficit during 2002–2003 and a delayed start of the rainy season in 2003–2004 [45].

Heat waves cannot be considered causes of drought, not least because of their usual shorter duration. While heat waves last from a few days to a few weeks, droughts can last for several years [61,65]. Drought favors the occurrence of heat waves by reducing soil moisture and, consequently, its ability to absorb energy. On the other hand, heat waves intensify droughts by promoting additional evaporation of soils and living beings and can promote short droughts, from a few days to two months, in SA [61,64].

### 3.2.2. Drought Indices

Of the total number of studies used in the systematic review, 33 (62%) documents used indices (Table 5), namely 28 (53%) used drought indices, 11 (21%) vegetation indices, and 7 (13%) climate indices. The most used indices were the SPI (28%) and the SPEI (25%), followed by VCI and NDVI (9%), the Standardized Runoff Index-SRI (8%), ENSO (8%), and Southern Oscillation Index (6%).

Although less frequently, other indices were also used in the documents selected for the literature review, namely: Base Flow Index [66], Leaf Area Index [67,68], Standardized Soil Moisture Index [24,47], Antarctic Oscillation, Indian Ocean Dipole, North Atlantic Oscillation, Quasi-Biennial Oscillation, Tropical North Atlantic, Tropical South Atlantic, Sunspot Count [55], Soil Moisture Deficit Index, Evapotranspiration Deficit Index, Root Stress Anomaly Index, Groundwater Resource Index [69], Southern African Rainfall Index [47], Drought Excess Probability Index [43], Temperature Condition Index [70,71], Global Vegetation Index and PDSI. Although the PDSI is usually considered a good and globally used index, only one study included in the literature review used this index to study drought in SA [15,16].

**Table 5.** Most used indices of the studies. The absolute (N) and relative (n) number of citations are also provided.

| Indices | Studies | N (#) | n (%) |
|---|---|---|---|
| SPI | [24,29,33,41,45,48,51,56,58,71–76] | 15 | 28 |
| SPEI | [24,33,41,44,52,53,55–58,65,68,74] | 13 | 25 |
| SRI | [24,69,74] | 4 | 8 |
| ENSO | [50,54,55,77] | 4 | 8 |
| SOI | [47,48,56] | 3 | 6 |

A total of 17 (33%) documents used more than one index, which suggests that one single index may not be able to capture all aspects of the drought regime [56,66,69]. Some studies recommend the use of more than one type of index to study drought [24,44,68,78].

It is important to assess the use of drought indices as these time series/fields are used by researchers to determine drought descriptors (frequency, duration, severity, etc.) at different time and space scales. These indices aim to assess the local water balance and can be computed based on different climate, soil, and vegetation data. This approach allows historical climatological assessment of the drought regime, including the spatial and temporal distributions of the descriptors, as well as the monitoring of drought at shorter time scales. Knowledge of the spatial and temporal variabilities (intra and interannual) of drought regime descriptors is essential for a diverse audience and supports political decisions and drought management, including risk assessment and mapping and early warning systems [4,44,45].

### 3.2.3. Other Drought Assessment Methodologies

The literature review includes 20 (38%) studies that did not use drought, climate, or vegetation indices to study the drought regime in SA, but adopted other methodologies to study drought factors or impacts in SA, namely in biomes, hydrographic basins,

diseases, and food security, as well as case studies of specific drought events identified and characterized in previous studies [46,60,62,79–82]. The list of these methodologies includes (i) historical climatological analysis, including correlation and anomaly methods [43,56,58,78,79,81] (ii) mathematical/statistical methods and models, such as Wavelet analysis [55], maximum value composite technique and Mann–Kendal Tau correlation coefficient [52], Run–sum method [78], linear regression models, and non-linear methods [29,53], and extreme value theory [51], (iii) management tools [24,74,83], and (iv) atmospheric and hydrological physical models, including the use of global hydrological models and simulations [24,61,62,67,69,84], regional circulation models [33,72], global circulation models [62] or their simulations to estimate drought projections for future climate scenarios, such as the Phase 3 (CMIP3) and Phase 5 (CMIP5) Coupled Model Intercomparison Project [61,64].

It is important to discuss the methodologies identified in the literature review, namely with other methods, resources, and tools used in the assessment of the drought regime in other areas of study. This discussion includes the pros, cons, and potential application in the SA. Firstly, many other indices are often used to assess drought. For example, there are indices to assess drought in rivers [85] and new indices developed and applied, namely to assess drought trends in Europe, such as the Modified Rainfall Anomaly Index (MRAI), Water Balance Anomaly Index (WBAI), Aggregated Drought Evaluation index (ADEI), which, due to its application, can also be calibrated and applied in SA [86]. Some other methods to estimate drought characteristics, especially hydrological drought, are time series analysis/modeling, regionalization procedures to estimate/extrapolate low-flow/drought characteristics spatial distribution (simple estimation methods, multivariate analysis, regional regression models, hydrological mapping procedures), and frequency analysis using probability distribution analysis, extreme value analysis, regional frequency analysis, severity-area-frequency curves [85]. Following the recent and global trend of using machine learning methods, several researchers have proposed different algorithms for drought modeling, hazard monitoring, forecasting, and impacts, e.g., [87–91]. These methods have the advantage of not relying on prior knowledge of the phenomena/processes and being data-driven, i.e., the models are calibrated based on previous experiences [92]. In essence, these algorithms can apprehend, model, and simulate the relationship between predictors and predictand, based only on data from the independent (input) and dependent (output) variables [93]. There are a large number of these algorithms to implement classification/regression techniques (supervised learning) and clustering (unsupervised learning), which can be tested and used to study drought in SA [94–99].

Regarding additional resources to assess drought, there is now a vast set of observed databases (e.g., European Climate Assessment and Dataset project, Climate Research Unit, and MetOffice UK climate datasets), analysis and reanalysis (e.g., fifth generation ECMWF reanalysis—ERA5- the NASAS Modern-Era Retrospective analysis for Research and Applications—MERRA-, the Japanese reanalysis—JRA, the NOAA 20th-Century Reanalysis, the NCEP/NCAR Reanalysis—NCEP/NCAR). These datasets have several pros and cons. The data are available in different locations (e.g., meteorological stations) or a global network with relatively high spatial and temporal resolution. These databases are freely accessible and include time series or fields of climate elements (e.g., different types of precipitation, air temperatures, and humidity, wind), as well as other useful parameters in the study of drought (e.g., radiation, evaporation/potential evaporation, surface and subsurface runoff, soil moisture at various depths).

Nowadays, researchers also have access to a vast set of variables and parameters obtained by satellite remote sensing [100] that provides data at local, synoptic, or even global scales with (i) a coarse, high, or very high spatial resolution, depending on the size of the pixel which can vary from tens of meters to a few kilometers, (ii) high to very high temporal resolution, depending on the type of satellite polar or geostationary orbiting and with a frequency of overpass ranging from a minimum of every 3–4 days to a maximum of 92 or more, and (iii) high spectral resolution which corresponds to the number of spectral bands of the radiometer onboard and covers parts of the electromagnetic spectrum of

visible and infrared radiations reflected or emitted by the surface of the globe. In the last three decades, several satellite-derived vegetation indices have been developed to monitor drought from local to global scales, and, in addition, remote sensing data collected by several satellite-based instruments have also been used to estimate several crucial variables related to drought that include land surface temperature, evapotranspiration, soil moisture, and precipitation.

One of the widely used remote sensing programs is the open access Earth Data of the National Oceanic and Atmospheric Administration (NOAA) (https://modis.ornl.gov/, accessed on 27 June 2023) that provides data, since 2000, of the land surface obtained by the radiometer Moderate Resolution Imaging Spectroradiometer (MODIS) on-board polar-orbiting AQUA and TERRA satellites, and include vegetation indices (NDVI/EVI), Vegetation Continuous Fields, Thermal Anomalies and Fire, Surface Reflectance, Evapotranspiration, Leaf Area Index, Land Surface Temperature, Burned Area, Land Cover, etc. [101–111].

The European Meteorological Satellite Agency (EUMETSAT) (https://www.eumetsat.int/, accessed on 27 June 2023) is an intergovernmental organization with 30 member states and operates the: (i) Geostationary satellites Meteosat-10, and -11 over Europe and Africa, and Meteosat-9 over the Indian Ocean, (ii) two Metop polar-orbiting satellites as part of the Initial Joint Polar System shared with the NOAA, and (iii) Jason-3 and Jason-CS/Sentinel-6 as a partner in the cooperative sea level monitoring Jason missions involving Europe and the United States. EUMETSAT has created eight Satellite Application Facilities (SAFs) that represent dedicated centers for processing satellite data and form an integral part of the distributed EUMETSAT Application Ground Segment. One of these SAFs is the so-called Land Surface Analysis (LSA SAF), mainly centralized at Instituto Português do Mar e Atmosfera (IPMA) in Portugal and uses remotely sensed data for land, land–atmosphere interactions, and biosphere applications. The key focus of LSA SAF is to develop and process satellite products that characterize the continental surfaces, such as radiation products, vegetation, evapotranspiration, and wildfires. The main products developed by LSA SAF are Land Surface Albedo, Land Surface Temperature and Emissivity, Downward Shortwave Flux, Downward Longwave Flux, Net Longwave Flux, Leaf Area Index, Gross Primary Production, Normalized Difference Vegetation Index (NDVI), Evapotranspiration, Latent and Sensible Heat Fluxes, Fire Radiative Power, and Fire Risk Mapping [103,112–120,120–122].

Remote sensing data also have their pros and cons, but they can be very useful for assessing the drought regime in southern Africa, as they cover large regions in a relatively consistent and homogeneous way and provide reliable information with relatively high resolution space-time, for a wide range of elements and parameters. There is also a vast set of other databases, such as river networks and lakes, land cover and use, topography, etc. Nowadays, these databases have a long duration and high spatial and temporal resolution. These data are necessary and commonly used in the different methodologies used in the study of drought and can be used in the study of its regime in SA.

In addition to data, there is also a vast array of other resources available that can be used to study drought in SA, including publications, documents, videos, programs, projects, products, and services available on the websites of the World Meteorological Organization [123], United Nations Convention to Combat Desertification [123], Copernicus [123], US National Weather Service [124] or US Department of Agriculture. For example, the European Drought Observatory [125] portal provides drought information, graphs, and time series at the European level as well as a tool to compare several indices, including SPI, Standardized Snowpack Index (SSPI), Soil Moisture Anomaly (SMA), Anomaly of Vegetation Condition (FAPAR Anomaly), Low-Flow Index (LFI), Heat and Cold Wave Index (HCWI), Combined Drought Indicator (CDI). Copernicus also has a web-based tool for assessing the impacts of drought on water resources.

Eventually, one of the most useful resources is the drought monitors. The EDO has a large set of drought mapping tools, including an Interactive Mapviewer, to produce and see maps and check indices and other tools to, for example, download and analyze

data, and get and compare time series of different drought indicators. The NWS provides maps of the U.S. Drought Monitor (which is a weekly product that provides a general summary of current drought conditions), precipitation, maximum air temperature, Multi-Indicator Drought Index (MIDI), PDSI, Evaporative Demand Drought Index (EDDI), and the Seasonal Drought Outlook, which foresee the drought tendency. In this respect, it is worth mentioning the SPEI Global Drought Monitor (https://spei.csic.es/map, accessed on 28 June 2023), which provides near real-time information about drought conditions. This monitor has the advantage of providing time series and maps at the global scale of SPEI at time scales from 1 to 48 months, but at a spatial resolution of 1 degree. The Global Drought Monitor website also provides a global 1-degree gridded SPEI dataset for the period 1901–2011 and software tools (SPEI R package and auxiliary functions) to compute and analyze SPEI time series under various data scenarios. A new version of the global drought monitoring system based on ERA5 reanalysis provides near real-time information at a 0.5° resolution updated weekly [126].

The UNCCD has a Drought Toolbox (https://www.unccd.int/land-and-life/drought/toolbox, accessed on 28 June 2023), which "provides tools, case studies, and other resources to support the design of the National Drought Policy Plan to boost the resilience of people and ecosystems to drought". The Drought Toolbox is organized into three modules/pillars: Monitoring and early warning, Vulnerability and risk assessment, and Risk mitigation measures. The UNCCD supported countries in designing national drought plans, and it is worth mentioning that in 2020, a few SA countries, including Eswatini and Zimbabwe, defined their National Drought Plans. This shows that all these resources can be used or developed to study, monitor, and manage drought in SA, including data, indices, mapping, and analysis tools.

### 3.2.4. Drought Descriptors

The documents included in the literature review describe studies on meteorological, hydrological, and agricultural droughts. Only one document describes a drought study from a 1-month to 24-month scale, every three months [68]. Additionally, this study focused on the impacts of drought on vegetation in biomes in particular. We identified 13 (25%) studies assessing the spatial distribution, e.g., [45,51–53,68,70,80], 8 (15%) evaluated the intra-annual variability, e.g., [45,48,53,58,69], and 19 (36%) estimated the interannual variability of drought descriptors, e.g., [33,47,52,53,55,68,84,127]. We also found that 17 (32%) studies assessed the frequency of drought [24,45,61,69,76,80,127]. It was possible to identify in the documents that the assessed drought classes were severe/extreme drought with 15 (28%) studies [44,45,51,68,70,76,127] and 7 (13%) for severe/moderate drought [24,33,53,55,58,70,128]. However, none of the documents reports a complete study of the drought regime or the spatial and temporal distribution of all drought descriptors. For example, only five (9%) documents mention the drought duration, but in specific types of drought, study areas, or drought events [69,76,80,84].

### 3.2.5. Current Drought Regime
Number/Frequency

The analyzed documents provide information on the number or frequency of droughts in SA. The values vary with the period and area of analysis. Some studies suggest an average frequency of one drought every three to five years in the period from 1980 to 2007 [74,79,129]. In a study that investigated the response of the Leaf Area Index to drought in SA, the authors analyzed the drought in the period 1982–2011 and concluded an average number of 58 [68]. A similar number of events (ranging from 41 to 71) of droughts were estimated at all scales (from 1 to 24 months), although with a relatively higher frequency in the second decade (1992–2001) [68]. Results for specific countries are relatively different. In Zimbabwe, during the 1901–2000 period, there were 7 extreme, 3 severe, 9 moderate, and 12 mild droughts [48]. Meteorological droughts were more frequent in the Upper Kafue watershed compared to hydrological and agricultural droughts from 1984 to 2013 [24]. In

the study that developed the Blended Drought Index (BDI), which is an integrated tool for estimating the impacts of meteorological and agricultural drought as a climate-induced hazard in the semi-arid Cuvelai-Basin of Angola and Namibia [44], the SPI, SPEI, SSI, and VCI indices were used and assessed. The obtained results depict different spatial patterns of drought frequency for each index. The frequency of occurrence is higher in the southeast when using SPI and SPEI, in the southwest for VCI, and central/north for SSI.

Some studies report trends in the number/frequency of droughts, with the highest values occurring in the more recent decades [52]. Authors report a significant increase in the frequency of drought on a 24-month scale since the 1970s [45] and a higher frequency of droughts related to El Niño events [56]. For example, from the 1970s to 2016, the frequency of decadal droughts at 3- to 12-months scale increased in SA [55], and on average, they occurred in conjunction with heatwave events [61]. Although most of the studies reviewed do not cover the whole study area (SA), only part of it, and generally focus on South Africa and never on Lesotho, Eswatini, Malawi, and Botswana.

Duration

The literature revealed that the drought of the early 1990s was the longest, even when compared to the extreme drought event of 2015–2016 [56,69,76]. The extreme agricultural drought of 1991 only ended in early 1993, but the effects of the hydrological drought only ended in late 1993 [69]. In South Africa, the average duration of agricultural and hydrological droughts was longer when compared to meteorological droughts in the Upper Kafue Watershed/SA from 1984 to 2013 [24]. In the Berg River alone, in the Mediterranean part of South Africa, about 40% of the length of dry days occurred in the rainy season (December, January, and February: DJF) of 1994–1995 and 2015 to 2017 [84].

There are also some studies reporting trends in drought duration. For example, in the 1961–2016 period, the occurrence of long-lasting droughts varied from 20% to 68% in Zambia, and the duration of the rainy season tends to decrease since 1991, with a delay of (3 to 7) weeks from the beginning of the rainy season or which brings the onset of the dry season forward by (1 to 7) weeks [80]. The lack of information on the duration of the drought suggests the need for further studies focusing on this feature/descriptor of the drought.

Severity

The most intense drought in the 116-year historical record (from 1900 to 2016), which occurred in the period between October 2015 and March 2016, is also considered to be the most meteorologically severe since the 1980s [29,45,48,51–53,67,76,84]. The years 2001–2002, 2002–2003, and 2003–2004 experienced severe droughts at various scales [45,129]. The six strongest droughts on the two-year scale occurred during the last two decades and increased in severity and extent [45,68]. Since 1970 there has been an increase in the severity of severe droughts in the SA river basins (Orange, Limpopo, Zambezi, and Okavango) [33,69,74]. From 1980 to 2010, mild droughts in the southwest and northwest of the Cuvelai River basin in Angola and Namibia were also recorded [44]. Drought impacts on the biomes and vegetation of SA were recorded over the 20 years from 1998 to 2017 [48,50,68,83] and they proved to be severe in the semi-desert areas of Angola, South Africa, Mozambique, and Zambia, exacerbating plant stress [58] and increasing aridity by 11% between 1980 and 2007 [55]. The literature does present some data on drought severity but never at all timescales and rarely for the entire SA.

Spatial Extension

The included studies describe a considerable interdecadal variability in the spatial extent of drought since the early 20th century in Zimbabwe, Lesotho, South Africa, Eswatini, Mozambique, Southern Zambia, Botswana, Namibia, and part of Angola [45,51]. On the other hand, since 1970 there has been an increase in the extension of droughts in the major river basins of SA, specifically Orange, Limpopo, Zambezi, and Okavango [33,69], in the

river basins of the Incomati that crosses Eswatini, South Africa, and Mozambique [76], in the Cuvelai watershed, in Angola and Namibia since 1980 [44]. There is also an increase in the drought-affected area since 1903 in the biomes of the southwestern part of SA, especially rangelands and crops [52,53,68]. Based on the studies included in the review, it was not possible to assess the total extent of the area affected by all types of drought in all countries of SA, so this knowledge gap opens up new future challenges for the complete study of the spatial extent of drought in SA.

Lastly, in the previously mentioned study that developed the BDI, which uses a copula function to combine the SPI, SPEI, SSI, and VCI indices, drought frequency, duration, severity and, therefore, the spatial extent of the drought-affected area in the semi-arid Cuvelai-Basin of Angola and Namibia were assessed [44]. Preliminary results, obtained with each of the drought indices results, depict different spatial patterns for each index and drought descriptor, which suggests that the evaluation of the drought regime depends on the index used to assess drought and may require the combination of more than one drought index.

### 3.2.6. Drought Impacts

The literature review identified seven major types of drought impacts in SA: (i) Scarcity of potable water, food insecurity, and hunger [29,33,43,44,46,49,51,52,60,62,63,67,79,128–130], (ii) increases in malnutrition, morbidity, and mortality [29,50,52,60,72,76,130], (iii) loss of agricultural production [33,44,50–52,76,84], (iv) reduction of industrial and hydro-electric energy production [33,62], (v) pressure on the economy and promotion of emigration [33,60,79,84], (vi) regional humanitarian crisis [129], (vii) risk of groundwater drought [83], and (viii) degradation of ecosystems [60,63,68,129,131]. Drought is indeed one of the natural hazards that affect various sectors in SA, where many socioeconomic activities depend on rainfed agriculture [65,80].

The impacts of drought depend on its intensity, duration, and preparedness. SA has been characterized by strong interannual precipitation variability [45], and drought is considered by many authors to be a regular and recurrent feature of the types of climate of the region, as recurrent droughts continue to impact rural livelihoods and degrade the climate environment [74]. For example, from 1900 to 2013, about 870 thousand people died, and 414 million people were affected by drought in SA [129]. The extreme El Niño event of 2015–2016 caused a high rainfall deficit, which led to a major food crisis, severe hydroelectric power shortages, shortages of potable water, reduced harvests and livestock, conflicts, and access to water [62]. The same El Niño event resulted in more than half a million cases of acute malnutrition in children, 3.2 million children with reduced availability of drinking water, and increased infant mortality of children under five years, especially with the worsening of the drought in Angola, Malawi, Mozambique, Namibia, and Zambia [76]. Women's vulnerability to contracting diseases, generating public health crises, including the human immunodeficiency virus, is also promoted during drought events [29,43,51].

The Impacts associated with droughts are evident, and the expected population increase will increase the pressure on water resources for consumption and human activities, including farming production, energy, industrial, and service sectors, as well as natural ecosystems. For these reasons, it is necessary to deepen the scientific knowledge about the regime of drought in SA under current and future climate conditions.

### 3.2.7. Future Drought Regime

The drought projection studies in SA indicate that by 2050, air temperatures will be significantly higher, and there will be a reduction in precipitation, which will lead to the worsening of drought descriptors/regime [50,60]. Future projections indicate changes in oceanic, atmospheric, and climatic processes that will lead to an increase in temperature variability in the region, as well as a reduction in the amount of precipitation and duration of rainy seasons. Some of these studies also suggest a large increase in sudden droughts in semi-humid and semi-arid regions, which will reduce food production in these regions [61],

while the dry areas will become even drier [129]. Projections from the CMIP3 suggest an increasing trend of droughts during the summer seasons, from December to February [64]. Increased global warming levels (GWL) will differentially intensify the frequency, severity, and spatial extent of drought in different regions of SA [45,64,66,129] as well as in the watersheds of the main rivers (Orange, Limpopo, Zambezi, and Okavango) of the region [24,33]. In an assessment of future groundwater drought risk in the Southern African Development Community (SADC countries), the authors compare the projections for future (2080–2099) and reference (1989–2008) periods [83] and conclude an average increase of 36.4% in population and of 18.5% in the area affected by very high groundwater drought risk [83]. Findings are even more impressive for SA-specific countries. For example, in the reference period, only three countries have more than 5% of the country area at very high groundwater drought risk (Malawi 9.8%, Lesotho 6.6%, and Zimbabwe 5.3%), but the estimated increase in the area is greater than 40% in Zimbabwe (66.1%), Malawi (56%), and Mozambique (41.2%) [83]. Results for the population with very high groundwater drought risk are even more significant. In the reference period, three countries have more than 30% of the population at risk (South Africa 37.5%, Malawi 31.9%, and Zimbabwe 31.8%), but the projected increase is greater than 50% in Malawi (64.5%), Mozambique (63.5%), and Zimbabwe (53.9%) [83]. These projections of population and area increase in very high groundwater drought risk are particularly important, as they suggest, on the one hand, an important worsening of dry conditions in future climate conditions and, on the other hand, because they refer to hydrological drought imply increases, at least identical, in the risk of meteorological and agricultural droughts.

### 3.2.8. Study Limitations and Final Considerations

It is important to mention the limitations of the literature review results in the sense that these results depend on the selected and analyzed documents. Although the research equation was defined in the most general way possible to achieve the main goal of evaluating the state of the art on the drought regime in SA, some documents may not have been identified for different reasons. For example, the authors may not have referred to the study area or have used concepts or expressions that are synonymous or associated with drought, such as low flow, low moisture, dryness, water deficit, shortage, or scarcity. The inclusion and exclusion criteria were the most obvious, usual, and adequate, and a low number of documents were excluded to avoid introducing bias. More recently published documents that could eventually contain relevant information were also excluded.

Our study cited a significant number (16) of review and systematic review articles. Most of these articles (11) were used in the introduction to present/describe concepts and definitions [9,41,132], characteristics [36], research themes, methods, indices, patterns, connections [34], impacts [35,38–40,132], and policy decisions/management [37] of droughts in general terms, that is, not necessarily and specifically for SA. We intend that this work may be of interest to practitioners and constitute an element of study for all researchers interested in starting to explore the drought regime anywhere in the world and especially in SA. This is why the introduction provides a vast set of fundamental information about drought, and Section 4 is concerned with presenting and discussing the results of the literature review on the drought regime in SA.

The remaining (5) review articles were identified in the literature search and used in the literature review [41,63,133–135], as all focused on AS, except one whose study area was the entire African continent. These articles used in the literature review focused on drought periods and regions most susceptible to drought (e.g., arid zones and drylands), as well as very drought-specific topics such as human response and adaptation [133], drought hazard and desertification management [63], the value of artificial ponds for aquatic insects [134], trade-offs for arid-zone birds [135], and societal implications of different types of drought under climate change [41].

It is important to emphasize that none of these review articles focused on the comprehensive description of the drought regime in the SA, the main aim of this study. We

examined these existing review articles on droughts to extract the necessary information for our purpose, seeking to avoid redundancies, find gaps in knowledge, and differentiate points where we seek to do better and further concerning the characterization of the drought regime in SA. Finally, we are confident that we have differentiated our contribution from other reviews, not only in terms of the study's region of interest but in terms of reflecting on and sharing insights into the existing literature. This review follows on from other reviews on drought but has the added value of prior knowledge, previous experience, and new ideas from the authors, as well as a discussion on the pros, cons, possible new applications, and perspectives, aiming to define the manuscript as an innovative review contribution, capable of generating new ideas and studies on drought in SA.

## 4. Conclusions

The bibliographical research carried out in the Web of Science and Scopus databases allowed the general objective of identifying 157 documents on the drought regime in SA to be fulfilled. The application of the PRISMA2020 methodology, including the usual inclusion and exclusion criteria suitable for the purpose and aiming to minimize/eliminate the risk of bias, made it possible to reduce this list to 53 documents on which the literature review was carried out. Most of these documents addressed drought mainly in just a few countries, such as South Africa (43%), Mozambique (25%), and Zimbabwe (21%). Only 38% of the 53 documents describe studies carried out for the entire SA but mostly restricted to a drought event, one or two drought descriptors, some drought classes and time scales, impacts, or the creation of monitoring tools.

The bibliographical review revealed that Ocean–Atmosphere interactions are the main factor of drought in SA, in particular, the ENSO is associated with more than 66% of the severe droughts that occurred in the region. Other factors, such as anthropic influence (including fires, gas emissions, and global warming) and heat waves, were mentioned in 17% of the documents.

The systematic review revealed that 62% of the studies used drought, vegetation, and climate indices, especially the SPI and SPEI, as well as atmospheric/climatic and hydrological models to characterize the drought regime. In general, the authors study only some of the drought descriptors (number/frequency, duration, severity, area affected by the drought), and the results vary according to the methodology used (e.g., drought index) and region of study. However, there is some unanimity as to the fact that the drought of the early 1990s was historically the longest and the drought of 2015–2016 was the most meteorologically extreme.

The impacts of drought depend on its intensity, duration, and preparedness of communities, but include scarcity of potable water, reduced agricultural and energy production, food insecurity, morbidity and mortality, socioeconomic pressure, and degradation of natural ecosystems. For the future, the studies project a general worsening of the regime of all types of drought. The increase of the GWL will intensify the frequency, intensity and severity of the drought in different regions, including in the main river basins of the region.

Although some studies analyze some drought descriptors with different methodologies, no document describes the complete characterization of the drought regime across the SA, namely, a detailed space-time analysis that includes the inter- and intra-annual distribution of the different types of drought on all time scales. This conclusion suggests and motivates the realization of studies that fill this knowledge gap, support water/drought monitoring and managers as well as support decision/policymakers.

**Author Contributions:** Conceptualization, M.G.P. and M.A.; methodology, M.G.P. and F.M.C.; software, F.M.C.; validation, F.M.C., M.A. and M.G.P.; formal analysis, F.M.C.; investigation, F.M.C. and M.G.P.; resources, M.G.P. and M.A.; data curation, F.M.C.; writing—original draft preparation, F.M.C.; writing—review and editing, M.G.P. and M.A.; visualization, F.M.C. and M.G.P.; supervision, M.G.P. and M.A.; project administration, M.G.P.; funding acquisition, M.G.P. All authors have read and agreed to the published version of the manuscript.

**Funding:** This work is supported by National Funds from FCT—Portuguese Foundation for Science and Technology, under the project UIDB/04033/2020.

**Data Availability Statement:** All data used in this study are freely accessible on the platforms of data providers, referred to in Sections 2 and 3. The list of publications identified by the bibliographic search in Web of Science (WoS) and Scopus are platforms accessed with the link provided in Table 3 but can be provided by the corresponding author upon request.

**Acknowledgments:** We wish to thank Web of Science (WoS) and Scopus for providing access to the databases. Comments from the anonymous referees and the editor are also gratefully appreciated.

**Conflicts of Interest:** The authors declare no conflict of interest. The funders had no role in the design of the study; in the collection, analyses, or interpretation of data; in the writing of the manuscript; or in the decision to publish the results.

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
