# Peer review of "The Drought Regime in Southern Africa: A Systematic Review"

_climate, doi:10.3390/cli11070147_

Round 1

Reviewer 1 Report

Dear authors

Your article is very good and will serve other researchers. It will help in teaching too.

I made a few comments, questions and corrections that I am sure you will correct in a few minutes, allowing the acceptance of your paper.

Actually, I only found and highlighted a few mistakes or unclear words (to me). Kindly check the annotated PDF.

Author Response

Answers to Reviewer #1

Dear authors

Your article is very good and will serve other researchers. It will help in teaching too.

I made a few comments, questions and corrections that I am sure you will correct in a few minutes, allowing the acceptance of your paper.

Answer: We are very grateful to the reviewer for the time and effort taken to revise our manuscript as well as for the kind words of recognition of the quality, pertinence and usefulness of our manuscript. We are also very grateful to the reviewer for the suggestions for changes that we are sure will improve the quality of our manuscript.

Comments on the Quality of English Language

Actually, I only found and highlighted a few mistakes or unclear words (to me). Kindly check the annotated PDF.

Answer: We check the annotated PDF and implemented all the suggestions of the reviewer.

Reviewer 2 Report

The publication is interesting (especially for readers from Southern Africa), but I have a few comments.

First, the definitions and literature review in Introduction is insufficient (in my opinion).

·  Where does the definition of hydrological drought come from (in lines 63-64) - there are many definitions of this stage (type) of drought, but in the current definition I don’t know what "average level" is meant - daily? monthly? annual?

· There are also some inaccuracies, e.g. in "Drought indices are time series of numerical values" (line 73) - maybe it should perhaps read: "Series of drought indices are time series of numerical values"? - drought indice is not a series, but an indicator value.

· It's true that the PDSI, SPI, SPEI indicators are often used, but there are many others that are often used. For example, river drought is often defined as e.g. in Tallaksen L.M., van Lanen H.A.J. (eds.) (2004) Hydrological Drought - Processes and Estimation Methods for Streamflow and Groundwater. Developments in Water Sciences 48. Elsevier, B.V., 580.

The second problem, however, is more serious: why was only the equation in the title of the articles "Drought*", "Southern", "Africa*" (line 197) selected for analysis? Very often, the authors do not write the word "drought" in the title, but, for example, the name of the indicator. Authors sometimes use the phrase "low flow" etc. when examining hydrological drought. In addition, drought is often tested not on the entire continent, but in individual countries - which has not been included here in this paper… Maybe it would be good to search not only titles but also keywords?

As a result, only 41 papers were reviewed. It seems to me that in the case of a review paper in such a reputable journal as Climate, this is very small number. I suggest authors look for more papers or publication in another journal.

Author Response

Answers to Reviewer 2

Comments and Suggestions for Authors

The publication is interesting (especially for readers from Southern Africa), but I have a few comments.

Answer: We are very grateful to the reviewer for the time and effort taken to revise our manuscript as well as for the suggestions for changes that we are sure will improve the quality of our manuscript. We present below a response to each of the reviewer's comments.

First, the definitions and literature review in Introduction is insufficient (in my opinion).

 Where does the definition of hydrological drought come from (in lines 63-64) - there are many definitions of this stage (type) of drought, but in the current definition I don’t know what "average level" is meant - daily? monthly? annual?

Answer: As explained in the manuscript, one of our objectives with this literature search and review is to produce a document that can serve as a study element for doctoral students and young researchers of the drought regime in SA. One of the possible ways to achieve this objective is to provide a description, in the introduction of the manuscript, on the one hand, correct and as profound/detailed as possible and, on the other hand, succinct as necessary and compatible with the size of a scientific paper. This is the reason why we decide to present only some of the fundamentals but not all of the definitions. The second way is the literature review on the drought regime in SA, presented and discussed in section 3.

Regarding the definition of hydrological drought, we agree with the reviewer that some clarification is needed. We have decided to provide multiple definitions of hydrological drought to try to clarify the concept and we may have run the risk of not being entirely clear. The value of an average of a time series is dependent on the period considered to compute the average. In the context of hydrological drought, it seemed obvious to us that the period would have to be long, much longer than a day or even a month. But we agree with the reviewer in the sense that in a scientific article, there should be no room for subjectivity or free interpretation by the reader. The definition of hydrological drought referred to by the reviewer is similar to that of Tallasksen and van Lanen (2004) who refer "On the other hand, when a prolonged deficiency of precipitation occurs, it can trigger a hydrological drought and consequently cause a reduction in the normal levels of soil moisture, water flow, lakes, and groundwater in a given region". We understood, possibly wrongly, that using the word "normal", which for us climatologists have the meaning assigned by the World Meteorological Organization, could be unclear and we replaced the “normal” word for “average” (and not mean, which a different statistical meaning) but with the same meaning of being for a long period, typically 30 years. We were also careful to include the word “significant”, in a clear allusion to the concept of statistical significance, which is necessary for these definitions. In this sense, we clarify/change the definition of hydrologic drought

There are also some inaccuracies, e.g. in "Drought indices are time series of numerical values" (line 73) - maybe it should perhaps read: "Series of drought indices are time series of numerical values"? - drought indice is not a series, but an indicator value.

Answer: We agree with the reviewer and change the manuscript accordingly.

  • It's true that the PDSI, SPI, SPEI indicators are often used, but there are many others that are often used. For example, river drought is often defined as e.g. in Tallaksen L.M., van Lanen H.A.J. (eds.) (2004) Hydrological Drought - Processes and Estimation Methods for Streamflow and Groundwater. Developments in Water Sciences 48. Elsevier, B.V., 580.

Answer: We agree with the reviewer. We mention many other drought indices, especially in the new/revised version of the manuscript. We also include a reference to the indices used to asses river drought.

The second problem, however, is more serious: why was only the equation in the title of the articles "Drought*", "Southern", "Africa*" (line 197) selected for analysis? Very often, the authors do not write the word "drought" in the title, but, for example, the name of the indicator. Authors sometimes use the phrase "low flow" etc. when examining hydrological drought. In addition, drought is often tested not on the entire continent, but in individual countries - which has not been included here in this paper… Maybe it would be good to search not only titles but also keywords?

As a result, only 41 papers were reviewed. It seems to me that in the case of a review paper in such a reputable journal as Climate, this is very small number. I suggest authors look for more papers or publication in another journal.

Answer: In summary, we agree with the reviewer and the new version of the manuscript, the literature revision is formally based on 53 documents (which corresponds to an increase of about 30% from the previous 41) but is based on a much higher number of references. We performed a systematic literature search in WoS and Scopus, as explained in the manuscript; we did not look for documents/papers in any specific journal. We decide to use what seems to us the most general search equation with just 3 words aiming to identify all pertinent documents. Searching in other parts of the documents leads to a higher number but not pertinent documents. In the new version of the manuscript, we decide to include letters, systematic review papers, articles published before 2000 (excluded in the first version) and a few more papers selected by citation to increase the number of documents to review the literature. It is worth noting that, in recent cases, the number of papers used to review the literature is much small. In one of the review papers (Ruanza et al., 2022), the number of studies used in the review is 18, in the other (Militão et al., 2022) the literature review was based on 14 articles.

It is important to (almost) repeat here part of the answer to the academic review. We want to inform you that this manuscript does not describe an isolated work, but it is part of a research project whose main objective is to study in-depth and in detail the drought regime in Southern Africa. In this context, we adopted a broad/comprehensive definition of drought regime, which extends from the factors to the descriptors and consequences. This manuscript is the milestone of the first phase of the work plan, which consists of bibliographical research and literature review, aiming to know the state of the art and identify gaps in knowledge, to be explored in the next phases. For this reason, we aimed to find as many documents as possible on drought in southern Africa and therefore defined a very general/comprehensive search equation. However, regrettably, the number of manuscripts on the topic is not very high.

Round 2

Reviewer 2 Report

Thanks to the Authors for improving the manuscript.